# Public Health Implications of Cannabis Legalization: An Exploration of Adolescent Use and Evidence-Based Interventions

**DOI:** 10.3390/ijerph19063336

**Published:** 2022-03-11

**Authors:** Joseph Donnelly, Michael Young, Brenda Marshall, Michael L. Hecht, Elena Saldutti

**Affiliations:** 1Department of Public Health, Montclair State University, Upper Montclair, NJ 07043, USA; salduttie@montclair.edu; 2Center for Evidence-Based Programming, South Padre Island, TX 78597, USA; evidence_based@yahoo.com; 3Department of Nursing, William Paterson University, Wayne, NJ 07470, USA; marshallb3@wpunj.edu; 4REAL Prevention LLC, Clifton, NJ 07013, USA; hechtpsu@gmail.com

**Keywords:** adolescent cannabis use, recreational marijuana legalization, cannabis and mental health, cannabis and health implications, evidence-based interventions

## Abstract

This article examines the relaxation of state marijuana laws, changes in adolescent use of marijuana, and implications for drug education. Under federal law, use of marijuana remains illegal. In spite of this federal legislation, as of 1 June 2021, 36 states, four territories and the District of Columbia have enacted medical marijuana laws. There are 17 states, two territories and the District of Columbia that have also passed recreational marijuana laws. One of the concerns regarding the enactment of legislation that has increased access to marijuana is the possibility of increased adolescent use of marijuana. While there are documented benefits of marijuana use for certain medical conditions, we know that marijuana use by young people can interfere with brain development, so increased marijuana use by adolescents raises legitimate health concerns. A review of results from national survey data, including CDC’s YRBS, Monitoring the Future, and the National Household Survey on Drug Use, allows us to document changes in marijuana use over time. Increased legal access to marijuana also has implications for educational programming. A “Reefer Madness” type educational approach no longer works (if it ever did). We explore various strategies, including prevention programs for education about marijuana, and make recommendations for health educators.

## 1. Introduction

Under federal law, the use of marijuana remains illegal. In spite of this federal legislation, as of 1 June 2021, 36 states, four territories and the District of Columbia have enacted medical marijuana laws. There are 17 states, two territories and the District of Columbia that have also passed recreational marijuana laws. One of the concerns regarding the enactment of legislation that has increased access to marijuana is the possibility of increased adolescent use of marijuana. This concern has been raised by parents, educators, researchers, public health professionals, and other community stakeholders. In states that have increased legal access to marijuana, has there been an increase in adolescent marijuana use? How has use impacted adolescents? From both a public policy standpoint and educational perspective, how might we best approach the issue of reducing adolescent marijuana use? In this commentary, we will briefly explore the scope of marijuana legalization, the impact recreational legalization has had on the adolescent population, and the national and international response.

It may be surprising for some to discover that at the federal level, marijuana is still considered a Schedule I Substance under the Controlled Substance Act [1]. However, in 2013, the US Department of Justice updated their marijuana enforcement policy in an effort to address the state legalization initiatives. The policy confirmed that marijuana remained an illegal drug, but states would continue to be given the authority to determine marijuana laws and enforcement [2]. It appears that the federal policy requires states to enact their own regulatory protocols concerning “production, distribution, and possession of marijuana” [2]. This raises the question as to why federal law continues to prohibit marijuana use. The year 2022 may be a pivotal one for marijuana legalization. Congress is set to discuss several marijuana initiatives, including decriminalizing marijuana possession and use, and removing marijuana from the Schedule I Substance listing [3].

If one wants to learn more about the frequency with which marijuana, or other drugs, are used in the United States, there are several easily accessible sources, including Monitoring the Future, CDC’s Youth Risk Behavior Surveillance System (YRBSS), and the National Survey on Drug Use and Health. If, however, researchers were to use these sources in an attempt to determine the impact of recreational legalization of marijuana on adolescent marijuana use, they would be hard pressed to find a consistent trend [4]. Variability also exists when comparing pre and post recreational legalization rates in those states that have legalized recreational marijuana. For example, some states actually show decreases in adolescent use post-recreational legalization, while other states report increased use in select populations (e.g., recurrent marijuana users [5], college students [6]). Additionally, it is important to note a major limitation involving the most recent data collection years–the coronavirus (COVID-19) pandemic. COVID-19 significantly affected data collection methods for the 2019–2020 annual report; no data were recorded from mid-March until September 2020 [4]. As such, the authors caution against using such information when determining outcomes. With widespread surges of COVID-19 brought on by the Delta and the Omicron variants, in 2021, and continuing into 2022, the pandemic may continue to be an obstacle to obtaining accurate data. This makes it extraordinarily challenging to consider the impact of recreational legalization on the adolescent population, as the best data we have may well not provide an accurate reflection of actual use. When one considers the eight states that moved to legalize recreational marijuana during 2020 and 2021, attempting to determine changes in use, simply by comparing numbers, before and after legalization, will not likely provide accurate results [4].

Recent, but pre COVID-19, research, indicated that marijuana legalization has had a minimal impact on adolescent drug use however [7]. While some people may assume that legalizing recreational marijuana will increase use of marijuana by adolescence, at this point, we simply do not know whether this is actually the case. It is never easy to collect accurate data concerning adolescent drug use, but conditions surrounding the pandemic, including increased adolescent social isolation and significant disruptions in data collection, create substantial limitations.

Research has, however, substantiated the negative impact that recreational marijuana has had on general public health, specifically increases in emergency room visits, motor vehicle crashes, and traffic fatalities. In Colorado, significant increases were reported in all of three of these measures when comparing numbers prior to recreational legalization to post-recreational legalization [8]. These findings suggest that, perhaps, it may be easier to identify indirect consequences of recreational marijuana legalization, including the immediate and longer-term impact of prevention program.

## 2. Impact of Legalization

Legalization of cannabis has had both intended and unintended consequences. Medical marijuana can have some positive benefits for some health conditions. The legalization of recreational marijuana has also had a positive impact on increasing state tax revenue and decreasing arrests for simple possession charges. Cannabis, as a product, also has negative health consequences that can have serious long-term effects when used by youth or abused by youth and adults. This section will examine the unintended, however not entirely unexpected, consequences that legalization of cannabis has on youth. Canada legalized use, possession and sale of recreational cannabis in 2018 [9]. Unlike the United States, Canada has a federal requirement that all cannabis products carry a warning message concerning THC (Frequent and prolonged use of cannabis containing THC can contribute to mental health problems over time. Daily or near-daily use increases the risk of dependence and may bring on or worsen disorders related to anxiety and depression). Canada also requires tamper-proof packaging that is child resistant [9]. This pre-emptive consideration for protecting youth reflects the understanding that brain response to drugs and toxins have great variability during the life span, especially where neuro-toxicity is concerned [10]. Despite these proactive interventions, Canada has witnessed increases in severe cannabis intoxication in pediatric patients since legalization, with the ingestion of cannabis edibles the strongest predictor of intensive care admissions [11].

### 2.1. Youth Substance Use

#### 2.1.1. Youth under 12 Years of Age

The National Poison Data System identified a rise in cannabis ingestion in children 0–6 years old, with over 70% of those cases in states with legalized recreational use. These are only the cases where the child has been brought to the hospital for treatment, so the general number of children in this age group ingesting cannabis is unknown [12]. In this age group, the researchers concluded that increases in access to cannabis edibles due to legalization was a contributing factor in the rise of cases [12].

#### 2.1.2. Adolescent Cannabis Use

Scheier and Griffin (2021) also examined the availability of cannabis to minors, specifically adolescents [13]. As reported in the 2020 Marijuana Research Report, adolescent marijuana use (i.e., 8th, 10th and 12th graders) peaked in the late 1990s and began to decline through the mid-2000s before leveling off [14]. In 2021, an estimated 7.1% of 8th graders, 17.3% of 10th graders, and 30.5% of 12th graders reported using marijuana in the past 12 months [15]. Additionally, the majority of 12th graders who used marijuana in the last year preferred vaping as their method of administration [16]. There is also evidence that cannabis users are also more likely to smoke tobacco cigarettes [17]. Increased use, especially seen in the ‘past thirty day’ category, correlates with the decreased perception of risk when using [13].

According to the U.S. Surgeon General, there is no safe amount of cannabis use for adolescents [17]. The general impact of cannabis use on the adolescent brain affects memory, decision-making, attention and motivation [17,18]. Early studies related to the impact of cannabis on personality development indicated that use of cannabis by adolescents is more attractive to those who have specific characteristics including isolation, social criticism, and alienation [19]. More recent studies have demonstrated that there is a relationship between impulsive, risk-seeking behaviors, identified as neurobehavioral disinhibition, and use of cannabis [13]. Research demonstrates that there are some youth who will be more physiologically and psychologically drawn to using cannabis, regardless of its legalization status. With legalization also comes normalization, which opens the door for usage by those who otherwise, if it were not legal, would be unlikely to participate.

Certainly not all adolescents who try marijuana will go on to be chronic users, defined as use at an early age with continued and increasing use over time. For those who, however, do become chronic users the impact can be dire. Cannabis Use Disorder (CUD), which develops in chronic users, increases the risk of lower self-expectations, lower life and work satisfaction, poor academic performance and places the youth at higher risk for developing other substance use disorders [20]. CUD results in poorer cognitions in the areas of memory and executive functioning, can result in changes in brain structure and functioning resulting in altered decision-making capacity [16].

There is an abundance of evidence from multiple studies, including systematic reviews, which demonstrates how cannabis use affects adolescent development of psychiatric disorders [21,22,23,24,25,26]. As indicated by Radhakrishnan et al. (2014), youth exposure to cannabinoids, which would include Spice and K2, underlies some of the transient psychiatric symptoms that mimic psychosis. The moderators of this response to cannabis and cannabinoids include genetics, family history, ACEs, and age of initial use. These correlations do not indicate causation; however, they do provide research with the red flag that identifies the use of cannabis as one of the components in the increasing identification of psychosis in youth. Additionally, in 2020, a comprehensive review of the literature demonstrated that “Prospective epidemiological studies have consistently demonstrated that cannabis use is associated with an increased risk of subsequently experiencing psychotic symptoms and developing schizophrenia-like psychosis (26). There is a dose–response effect revealing that as consumption of cannabis increases so do the adverse psychiatric effects [13]. Adolescent cannabis use was also positively correlated with an increased risk for psychosis; however, the correlation does not indicate any causation. This association between increased risk for psychotic events, psychosis, and relapsing psychosis for adolescent cannabis use has also been well documented in a number of studies [23]. The moderating variables for the development of psychosis are frequency and amount of cannabis use and the potency of the drug. Dosage and age of onset of use increases the risk of severe psychotic response, as does exposure to childhood trauma, identified as adverse childhood experiences (ACES). On a positive note, studies have demonstrated that abstinence from cannabis use, as short as three months, can bring the youth back to healthy levels of brain functioning [18].

All youth, especially adolescents, are at risk for negative outcomes from cannabis use. This may be due to the important neuromaturation that occurs in adolescence, particularly in the area of executive function (prefrontal networks). At present, projections of effects of cannabis on the adolescent brain are based on older data, when levels of use were lower and potency was less. New strains of cannabis, combined with the current favorite route of administration (vaping), may impact brain development even further. Children with emotional challenges are at higher risk of substance use for self-medication, which then increases their likelihood of developing a psychiatric disorder. Additional research is needed to better understand the effects of marijuana use on adolescents, as well as the effects of legalization on adolescent use.

In the meantime, what can public policy makers, educators, parents, and public health professionals to educate young people about the risks of marijuana use? What prevention approaches seem to have the most impact?

### 2.2. Prevention

Legalization of recreational and medical marijuana has had several consequences as documented in this paper. However, to date, little is known about the implications of these changes on substance use prevention and particularly the focus on marijuana in those efforts. As prevention specialists work to find ways to deal with the legalization of recreational marijuana, it may be worth noting that in all states recreational use of both alcohol and tobacco is legal. Should education to reduce the health risks of marijuana differ markedly from education to reduce the health risks of alcohol and tobacco?

It is beyond the scope of this article to examine all drug prevention interventions. Instead, we have chosen to focus on two programs. The first is a take-home parent–child program, one that promotes parent engagement. The second program is a school-based, classroom intervention. Both programs are theory-based. Keep A Clear Mind, the parent engagement program makes use of Social Norms Theory, the Health Belief Model, and the Theory of Planned Behavior. Keepin’ it REAL, the classroom-based intervention makes use of Social Emotional Learning Theory and Narrative Engagement Theory. It is unclear at this time which behavioral theories will be most helpful to curriculum developers in developing effective prevention programs for adolescents. It is clear that these two programs have made use of different behavior theories, but have both produced positive results

#### 2.2.1. Parent Engagement

Keep A Clear Mind [27] is a parent–child, take-home program in drug education. The program has received a number of awards and recognitions and is listed on the National Registry of Evidence-Based Programs and Practices. The program includes four student activity booklets (alcohol, tobacco, marijuana, choices), four student incentives, and five parent newsletters. Students take the activity books home, one per week, and do the program with their parents. This largely involves reading material together and answering simple questions. Students receive a small incentive (bumper sticker, book mark, etc.) for showing their teacher that their parents have signed indicating they have worked with them to complete the activity booklet. After four weeks of activity booklets, the newsletters are sent home, again, one per week (or one every other week). The program is easy to use and because the program is done at home, it takes very little classroom time.

The sections of the “We choose not to use Marijuana” activity booklet, like the alcohol and tobacco activity booklets, include Let’s Talk About (in this case-marijuana), And that’s a Fact, Why do people choose not to use, Think for yourself, and a Contract to Think for Yourself (about marijuana). In the Let’s Talk About section, factual information is provided about marijuana, including information about legalization. This section acknowledges that more than half of the states in the U.S. have made some legal provision for the medical use of marijuana and a number of states have made recreational use of marijuana legal. Like alcohol and tobacco, even in states where recreational use is legal, it is only legal for adults. In this section, the program also reminds the readers that by federal law, marijuana use is illegal for everyone, even if used strictly for medical purposes.

Prevention specialists understand that information/knowledge is a necessary, but insufficient precursor to behavior change. Thus, the Keep A Clear Mind prevention strategy for alcohol, tobacco, and marijuana is to present material in the context of health behavior theory. For example, social norms theory [28] suggests that our behavior is influenced by our perceptions or misperceptions of these norms. If we think everyone is doing “it,” regardless of what that may be, we are more likely to do it ourselves.

Keep A Clear Mind also makes use of the Health Belief Model [29], presenting information about how marijuana affects the body. This includes some effects that potentially are quite serious (severity) and which can and do occur among adolescent users (susceptibility). The program also helps young people understand that the benefits of choosing not to use marijuana far outweigh any perceived benefits and real risks of using (risks/benefits). Finally, Keep A Clear Mind helps young people learn to say “no” and gives them practice in doing so (self-efficacy).

The Theory of Planned Behavior is based on the concepts of intention and perceived behavior control. A person who has the intention to engage in a behavior is more likely to actually engage in the behavior than someone who does not have that intention. Intentions are influenced by attitude towards the behavior and subjective norms. If children value what their parents think, and they believe their parents clearly do not want them to use marijuana, then the children are less likely to have the intention to use marijuana. Because Keep A Clear Mind involves children and parents working through and discussing material together, it gives parents a real opportunity to let their children know how they feel about the use of marijuana. Again, Keep A Clear Mind makes it clear that the vast majority of people do not smoke marijuana.

Perceived behavioral control refers to one’s perception of control over their behavior. It is assumed that this concept is reflective of the obstacles one has encountered in past behavioral performances. That is, people with higher perceived control are more likely to form intentions to perform a particular action than people who perceive they have little or no control. Keep A Clear Mind walks young people through the steps to saying no and gives them opportunities to practice saying no. The idea here is to enhance self-efficacy and create higher perceived control.

How is the program’s approach to marijuana different from its approach to alcohol and tobacco? Keep A Clear Mind indicates that alcohol and tobacco are drugs that are legal for adults to use. It also indicates that while under federal law, marijuana is an illegal drug, in some states, provisions have been made for medical use, and for adults, recreational use. Other than the mention of these differences related to legal status, there is little difference in approach to prevention across the three drugs. Keep A Clear Mind is available from the Center for Evidence-Based Programming. The web address is www.keepaclearmind.com.

#### 2.2.2. Classroom Intervention

D.A.R.E. (Drug Abuse Resistance Education), the largest school-based substance use prevention program in the U.S. Initiated in Los Angeles in the mid-1980s, D.A.R.E. once had a footprint in over 90% of the schools in the U.S. Following evaluations that did not support its efficacy, this footprint shrank noticeably, although it remained the largest such program. In response, D.A.R.E. made the determination that it was a dissemination vehicle rather than a curriculum and sought an evidence-based program to fill the void. Turning to information sources such as the National Registry of Evidence-based programs and practices (NREPP), now largely defunct, D.A.R.E. reviewed interventions that were deemed “model programs”, NREPP’s highest designation, and chose keepin’ it REAL due to its strong outcome evaluation evidence and a multicultural strategy that fit a national program. It was eventually endorsed in the Surgeon General’s report on addiction and found to have a $72:1 cost–benefit ratio in an independent evaluation.

The original keepin’ it REAL (kiR) was developed for implementation by teachers in middle schools using narrative and social emotional learning frameworks that stressed a highly interactive lesson plan. Social emotional learning theory (SEL) is premised on the idea that if youth develop strong, basic competencies they will be less likely to engage in risky and unhealthy behaviors such as substance use [30]. From this perspective, there is no need to focus on specific risk behaviors such as substance use since the competencies apply to all risks. This, of course, means that specific marijuana content is not obligated.

The premise of the companion narrative approach, derived from Narrative Engagement Theory, is that engaging stories (i.e., meet the criteria of realism, interest, and identification) provide mental and behavior models that re-story or change the narrative about a topic, in this case substances. Strategically, this involved performances of indigenous narratives about the SEL competencies that presented a drug-free life as fun and normative and resisting offers of drugs as communicatively and relationally competent. Here, introducing stories about marijuana would be useful; however, in a changing legal and resulting social environment presenting stories that are static (e.g., those in videos or written form) is problematic because the narrative is changing rapidly.

It should be noted that neither approach emphasizes drug information and its companion fear appeals (i.e., scaring youth not to use drugs), which had been the main strategy of many prior prevention interventions, and which had not proved to be an effective strategy in several meta-analyses. Instead, drug “facts” were used in the lesson on risks and consequences.

When D.A.R.E. licensed kiR in the mid-2000s, they were onboard with these approaches, although it was agreed that the lessons had to be “DARE-ified” to adapt to delivery by police officers. While D.A.R.E. provides an extensive, 80-h training, the officers are not classroom teachers and, as a result, require more explicit instructions (i.e., they cannot simply be told to “lead a discussion”). During this process, the original kiR premise of presenting drug facts/information only in the context of the lesson on risks and consequences and only about alcohol and tobacco came under question. Since the previous D.A.R.E. curricula had placed much heavier emphasis on information (and maybe fear?) and D.A.R.E. serves multiple constituencies, some of which maintain a belief in drug information and fear tactics, this proved challenging. Others raised the valid point that marijuana was prevalent in their communities and the students they taught would want to know about it. The developers were told that the officers needed to be prepared to respond to questions about marijuana that they were likely to face. A work group on the topic was convened and created a “discussion guide” for officers to use to address marijuana, a topic that was likely to become increasingly relevant under widespread legalization of cannabis throughout the U.S, both Medical and Recreational legalization. The strategy that was developed was to treat marijuana like any other topic that might come up in discussions of risks and consequences by using questions to focus students to apply what they had learned to this substance. The officers are told:

“If students introduce the subject of marijuana, not only does this satisfy the concern of age appropriateness, but it also serves as an indication that the ensuing discussion will have particular meaning to the students. It is proven to be more effective to discuss drugs, risks and consequences, decision-making, and resistance strategies when the students show an interest by initiating the discussion. As part of the D.A.R.E. kiR elementary curriculum, a discussion guide has been provided to D.A.R.E. officers for incorporation into lessons when appropriate. The marijuana discussion guide has been constructed so that it reflects the design of the D.A.R.E. kiR elementary lessons, when employed it integrates in a seamless fashion.”

The officers were then told to remind students about the definition of a drug and discuss if marijuana meets it. Then remind them about risk and consequences, again discussing the application to marijuana. Officers were provided with information they could use about the effects on the mind and body to facilitate this discussion.

The advantage of this approach is that it allowed a national program to respond in a way that adjusted to local circumstances, including legalization status and community norms. It also allowed class discussion to adapt content to the local culture—the stories or narrative that emerges localizes the curriculum. With the emergence of vaping, it also allows adjustment to different delivery mechanisms. Unfortunately, over the years changes in D.A.R.E.’s administration led to the abandonment of this discussion guide, leaving officers to fend for themselves. As with any large, national organization it is likely that a great deal of variation has emerged in how the topic is handled.

## 3. Conclusions

The emerging marijuana legalization landscape provides both challenges and opportunities for the prevention community. One hopes that norms and attitudes would not become overly positive, i.e., legalization will not be equated with safety or health. It seems clear that the potential profits for the cannabis industries, and the lure of increased tax revenue, will likely translate into even more states, and possibly the federal government, legalizing marijuana. The argument has often been made that marijuana is no worse than alcohol or tobacco. The research available today may not allow one to accurately quantify the relative risk of these three drugs. For arguments sake, however, say there is no difference in risk. That is not much of a recommendation. Remember, 88,000 people die each year in the U.S. from alcohol-related causes and tobacco is responsible for 480,000 deaths per year in the U.S. (and millions of deaths each year worldwide). Regardless of legal status, it is important to encourage young people to avoid using marijuana—and alcohol and tobacco. We should also encourage business and policy makers to look beyond profits and revenue streams in addressing legalization.

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
