# Peer review of "Public Health Implications of Cannabis Legalization: An Exploration of Adolescent Use and Evidence-Based Interventions"

_ijerph, 2022, doi:10.3390/ijerph19063336_

Round 1

Reviewer 1 Report

The topic tackled by this paper is important and there is likely a large audience of potential readers for a paper of this type.  The title and abstract promise an important discussion of use available data and prevention programming specific to the adolescent population.  At present however, the manuscript is somewhat hard to follow and could benefit from restructuring, a more streamlined discussion of use among adolescents, and a more thorough discussion related to prevention programs. Additionally, a thorough proofread is needed to correct errors such as incorrect program names and acronyms which add to challenges with clarity.

If the main goals of the paper are indeed to explore adolescent use and present evidence-based interventions, it would help to re-structure the manuscript into two overarching sections with these goals.  

As it stands, the introduction tackles a number of topics but doesn't provide a clear roadmap for the paper.  Moreover, at present, the introduction does not make any mention at all of prevention programming, why it is important and what types of programs show promise for cannabis use prevention among adolescents. 

It seems the key argument of the paper is as follows: Additional data are needed to understand the impacts of legalization on cannabis use among adolescents.  We do know that cannabis use poses a danger for adolescents and have some data to suggest that use will increase in association with legalization.  Given the established health risks, we can not afford to wait to have better data to act to address the issue of use among youth, particularly since legalization is becoming more widespread. Prevention programming aimed to stem use/increasing use among adolescents is needed.  Evidence based prevention programs are available, and there are some lessons from alcohol and tobacco prevention programming that can be applied to marijuana prevention programming. 

Given this, a restructuring to the following format would allow the reader to more easily follow the information presented:

Cannabis Use Among Adolescents

What dangers does marijuana use pose for adolescents?  These data are clear and easy to describe.

How widespread is legalization?  What do we know about the relationship between legalization and use?  How does this apply to adolescents?

What data do we have about marijuana use among adolescents over time, in states where it is and is not legal?   Note issues associated with accuracy of recent data (e.g., COVID pandemic)

Prevention Programs

What types of programs are likely most beneficial when it comes to prevention? 

What behavioral change theories show the most promise in prevention programming for adolescents?

The paper currently takes a deep dive into two specific programs, one school-based and one parent-focused.  A higher-level overview of the the body of evidence for various types of programs, including school-based programs, but also community-based, after-school, and parent-engagement focused programs is needed.   

The current program descriptions mention a number of health behavior theories. This information is important, but again it is hard to follow.  

With regard to proofreading/typos, a few examples follow:

Line 46: Please correct to read Controlled Substance Act (vs. Control)

Line 59: If is repeated "If, however, if"  (several other similar instances throughout the paper)

Cannabis Use Disorder (CUD) is introduced on line145.  The following text repeatedly references ACU.  Is this a typo and it should be CUD?  What is ACU?  

Lines 182-183 read: "Should education to reduce the health risks of marijuana differ markedly from education to reduce the health risks of marijuana?"  Presumably, the second marijuana should read alcohol and tobacco.

In sum, this manuscript would benefit greatly from restructuring,  presenting the data and arguments related to adolescent use more clearly and concisely and presenting additional information related to prevention programming.

The paper presents important information and has the potential to make an important contribution to the literature.  

Reviewer 2 Report

The article is well explained and deals with a topic of great interest today, the consumption and legalization of cannabis, mainly among young people.

The only thing I miss is the comparison of the data with other countries, both in terms of epidemiology and preventive strategies, in order to give a more holistic view of the topic.  

Reviewer 3 Report

The article by Donnelly and colleagues is well written and compares two different methods for educating adolescents about the use of marijuana.  This topic is particularly of interest because of the growing legalization of recreational and medical cannabis not only in the United States, but globally.  My major concern has to do with the hazards of cannabis, while it is true that cannabis is not without hazards, the author’s at times present a biased opinion or skew to the facts and research.  In particular:

  1. In section 2, lines 95-96, by comparing cannabis in this manner to tobacco the author’s make it sound like the long-term health consequences of the use of both drugs is similar. Long-term cannabis use can increase the risk of schizophrenia and lower IQ in adolescents, in some individuals, as the author’s point out later, these risks are also relatively low in most individuals.  From a health standpoint these issues are lower than the hazards of tobacco use, which include deadly diseases such as lung cancer, oral cancer, COPD, and emphysema, and at higher rates. 
  2. In section 2.1.2, lines 150-162 and 169-170, the authors fail to point out that these studies remain unclear if the increase in psychosis associated with cannabis is a result of cannabis use or if the use of cannabis is a result of the underlying psychosis of the individuals (where people are using cannabis to self-medicate their condition). This remains unresolved in the cannabis field.

 Minor/Technical Issues:

  1. Line 59, the second “if” is not needed
  2. Line 144 “however” should be removed
  3. Line 183 I believe “marijuana” at the end of the sentence should be “tobacco and alcohol”
  4. Line 220 “are” should be removed
  5. Line 280 “drugs facts” should be drug “facts”
  6. Line 290 there is a reference to Figure 1, but there is no figure in the provided manuscript

Overall, I think this paper is well organized and written, however (and I understand this is a commentary), should do a more accurate job presenting the hazards associated with marijuana, and in the in comparison of these hazards to those of tobacco and alcohol use.

Round 2

Reviewer 1 Report

The revisions greatly improve the ease with which the reader can understand important concepts presented by the authors, and the flow of the paper overall has been enhanced.  This is an important topic and it will be great to add this paper to the limited extant literature.

Reviewer 3 Report

Overall the author's have done an excellent job addressing my concerns, and I have no further issues with the manuscript.